# Isolation and Identification of an α-Galactosidase-Producing *Lactosphaera pasteurii* Strain and Its Enzymatic Expression Analysis

**DOI:** 10.3390/molecules27185942

**Published:** 2022-09-13

**Authors:** Yan Zhao, Jinghui Zhou, Shan Dai, Xiaozhu Liu, Xuewen Zhang

**Affiliations:** 1Key Laboratory of Crop Epigenetic Regulation and Development in Hunan Province, Changsha 410128, China; 2College of Bioscience and Biotechnology, Hunan Agricultural University, Changsha 410128, China; 3College of Horticulture, Hunan Agricultural University, Changsha 410128, China; 4College of Food & Pharmaceutical Engineering, Guizhou Institute of Technology, Guiyang 550000, China

**Keywords:** α-galactosidase, *Lactosphaera pasteurii*, expression analysis, gene clone

## Abstract

α-Galactosidase (EC 3.2.1.22) refers to a group of enzymes that hydrolyze oligosaccharides containing α-galactoside-banded glycosides, such as stachyose, raffinose, and verbascose. These enzymes also possess great potential for application in sugar production, and in the feed and pharmaceutical industries. In this study, a strain of *Lactosphaera pasteurii* (WHPC005) that produces α-galactosidase was identified from the soil of Western Hunan, China. It was determined that the optimal temperature and pH for this α-galactosidase were 45 °C and 5.5, respectively. The activity of α-galactosidase was inhibited by K^+^, Al^3+^, Fe^3+^, fructose, sucrose, lactose, galactose, SDS, EDTA, NaCl, and (NH_4_)_2_SO_4_, and enhanced by Ca^2+^, Fe^2+^, Mn^2^, Zn^2+^, glucose, and raffinose. The optimal inducer was raffinose, and the optimal induction concentration was 30 μmol/L. The α-galactosidase gene was cloned using random fragment cloning methods. Sequence analysis demonstrated that the open reading frame of the α-galactosidase gene was 1230 bp, which encodes a putative protein of 409 amino acids in length. Bioinformatics analysis showed that the isoelectric point and molecular weight of this α-galactosidase were 4.84 and 47.40 kD, respectively. Random coils, alpha helixes, and beta turns were observed in its secondary structure, and conserved regions were found in the tertiary structure of this α-galactosidase. Therefore, this α-galactosidase-producing bacterial strain has the potential for application in the feed industry.

## 1. Introduction

α-Galactosidase (EC 3.2.1.22) refers to a group of enzymes that hydrolyzes oligosaccharides containing α-galactoside-banded glycosides, such as stachyose, raffinose, and verbascose, and possesses great potential for application in sugar production and in the pharmaceutical and feed industries [1,2]. It was reported that approximately 1% of oligosaccharides in the most common feedstuffs and soybean meal are in the form of α-glycoside [3,4]. The oligosaccharides cannot be decomposed and utilized by livestock due to the lack of an enzyme in their digestive system [5]. The application of α-galactosidase in additive supplements will increase the utilization of these carbohydrates and reduce the flatus-producing factors in the feed. In addition, α-galactosidase can be applied to digest D-raffinose in sugar beet syrup to favor crystallization and increase the sucrose yield [6].

α-Galactosidases widely exist in many microorganisms, including prokaryotes and eukaryotes [7]. Many microbial α-galactosidases have been identified and purified from fungi, and they are suitable for industrial applications because of their extracellular localization, acidic pH optima, and broad stability profiles [8]. Many α-galactosidases have been purified and studied in detail [9]. Wang et al. [10] identified an extracellular α-galactosidase (Gal27A) with a high specific activity of 423 U/mg in thermophilic *Neosartorya fischeri* P1. This was also the first report on the purification and gene cloning of *Neosartorya* α-galactosidase. A protease-resistant α-galactosidase from *Pleurotus djamor* with broad pH stability and satisfactory hydrolytic activity toward raffinose family oligosaccharides was introduced by Hu et al. [8]. In addition, a novel α-galactosidase gene from *Penicillium* sp. F63 CGMCC 1669 was cloned, characterized, and expressed in *Pichia pastoris* [11]. However, research on the α-galactosidases and their enzymatic characteristics from *Lactosphaera pasteurii* remains rare.

Herein, a strain of *Lactosphaera pasteurii* that excretes α-galactosidase was identified, and its enzymatic characteristics were analyzed. Moreover, the α-galactosidase gene was further cloned via random fragment cloning methods, and its sequence was analyzed using bioinformatics software and databases. An enzymatic expression analysis of this α-galactosidase was also performed.

## 2. Materials and Methods

### 2.1. Isolation of α-Galactosidase-Producing Strains

Five different soil samples were collected from locations rich in soybean meal near a tofu factory located in Xiangxi autonomous prefecture of Hunan province, China. Samples were diluted to 10^−4^ to 10^−6^ with saline solution and then spread on preliminary screening agar plates (yeast extract 5 g/L, peptone 10 g/L, NaCl 10 g/L, agar 15 g/L, and 5-bromo-4-chloro-3-indolyl-α-D-galactopyranoside (X-α-gal) 20 mg/L, pH 7.0). All plates were cultured at 37 °C for 48 h, and the positive colonies were identified by their blue color.

### 2.2. α-Galactosidase Activity Assays

The positive clones were transferred into the secondary screening medium (yeast extract 5 g/L, peptone 10 g/L, NaCl 5 g/L, and raffinose 2.5 g/L, pH 7.0). *p*-Nitrophenyl-d-galactopyranoside (p-NPG, Sigma, St. Louis, MO, USA) was used as a substrate to measure the α-galactosidase activity. Briefly, cultures were collected via centrifugation at 4000 g for 5 min. The supernatant was used as an enzyme solution. Enzyme solution (0.1 mL) was mixed with p-NPG solution (0.2 mL, 0.002 M) in citrate phosphate buffer (0.1 M, pH 5.0). The reaction was maintained at 45 °C for 60 min, and stopped by adding Na_2_CO_3_ (2.0 mL, 0.25 M). The released pNP was measured spectrophotometrically at 400 nm. The amount of enzyme that released 1.0 μmol of p-NP from p-NPG per minute per milliliter was defined as one unit of the enzyme [12].

### 2.3. Identification of the Selected Strain

For the phenotypic characterization assay of the selected strain, cells were identified via Gram staining, recorded with a microscope (Olympus, Tokyo, Japan), and then classified according to their morphotype.

For the molecular identification of the selected strain, the 16S rRNA gene was amplified through the polymerase chain reaction (PCR) method with purified genomic DNA as a template. The PCR product was purified and then sequenced by a professional bio-sequencing company. A similarity search was performed using the GenBank databases. The specific primers used in this study are listed in Appendix A. The evolutionary analysis was carried out using Molecular Evolutionary Genetics Analysis version X software (MEGA X, The Biodesign Institute, Tempe, AZ, USA).

### 2.4. Effects of Temperature and pH on the Enzymatic Activity

To confirm the optimal temperature of the α-galactosidase for the selected strain, enzymatic activity was measured in disodium hydrogen phosphate and citric acid buffer (pH 5.5) ranging from 20 °C to 80 °C (20, 25, 30, 35, 40, 50, 55, 60, 65, 70, 75, and 80 °C) for 1 h.

To determine the optimal pH, enzymatic activity was measured under different pH values ranging from 2.6 to 8.0 (2.6, 3.0, 3.6, 4.0, 4.4, 5.2, 5.6, 6.0, 6.6, 7.2, and 8.0) at 45 °C for 1 h using p-NPG as the substrate.

### 2.5. Effects of Metal Ions and Chemical Reagents on the Enzymatic Activity

To investigate the influence of various metal ions and chemical reagents on the α-galactosidase enzymatic activity, the enzymatic activity was measured using metal ions (10, 5, and 1 mM) and chemical reagents (10, 50, and 100 mM) at 45 °C for 1 h with p-NPG as the substrate.

### 2.6. Effects of Glucose and Raffinose on the Enzymatic Activity

The cells of the selected strain were cultured in medium containing different concentrations of glucose or raffinose (10, 20, 30, 40, and 50 μmol/L) at 37 °C for 48 h. After treatment, the enzymatic activity was measured as described above.

### 2.7. Cloning of the α-Galactosidase Gene from the Selected Strain

Genomic DNA of the target strain was prepared using a DNA Extraction Kit (Shanghai, China) according to the manufacturer’s instructions, and partially digesting the DNA with *Sau*3A I into random fragments 3–5 kb in length (Thermo Fisher, Waltham, MA, USA). The pUC19 vector was treated with alkaline phosphatase and then digested with *Bam*H I (Thermo Fisher). The recombinant DNA molecules were constructed with the digested genomic DNA and pUC19. The ligation mixture was transferred into competent *E. coli* DH5α via electroporation. The target clones were identified through culture on Luria-Bertani (LB) medium containing 20 µg/mL 5-bromo-4-chloro-3-indolyl-α-D-galactopyranoside (X-α-gal) and 50 µg/mL ampicillin.

### 2.8. Bioinformatics Analysis of α-Galactosidase from the Selected Strain

The sequencing of the positive clones was completed by BGI (The Beijing Genomics Institute, Shenzhen, China). The nucleotide sequences were translated into amino acids using DNAMAN software (Lynnon Biosoft, San Ramon, CA, USA), and homology was compared using the online program BLAST (https://blast.ncbi.nlm.nih.gov/Blast (accessed on 1 August 2022)) from the National Center for Biotechnology Information (NCBI, Bethesda, MD, USA).

The isoelectric point and molecular weight were predicted using the Swiss Bioinformatics Research Center (http://www.expasy.org/tools/pi_tool.htmL (accessed on 1 August 2022)) protein analysis website. A prosite motif search was performed using protein online analysis software (http://www.predictprotein.org/ (accessed on 1 August 2022)). The secondary structure and tertiary structure prediction analyses were performed on the website of the International Laboratory for Biometrics and Evolutionary Biology and the Institute for Protein Biology and Chemistry (http://npsa-pbil.ibcp.fr/cgi-bin/npsa_automat.pl?page=/NPSA/npsa_dsc.htmL (accessed on 1 August 2022)). Transmembrane region prediction was conducted on the website of The Center for Biological Sequence Analysis (CBS) of the Technical University of Denmark (http://www.cbs.dtu.dk/services/TMHMM/ (accessed on 1 August 2022)).

### 2.9. Statistical Analysis

Excel 2010 software was used for data processing and plotting. Adobe Photoshop CS was applied to edit the figures. The results are expressed as the mean ± standard deviation.

## 3. Results and Analysis

### 3.1. Isolation of α-Galactosidase-Producing Strains

Eight isolates with high α-galactosidase production capacity were obtained via color reaction in preliminary selection medium containing 2 mg/mL X-α-gal substrate. The relative enzymatic activity of these colonies was further verified using the p-NPG method, and the isolate that produced the highest amount of α-galactosidase was WHPC005 (Figure 1). Therefore, this isolate was chosen for further analysis.

### 3.2. Identification of α-Galactosidase-Producing Strain WHPC005

First, Gram staining was performed to identify the characteristics of the α-galactosidase-producing strain WHPC005, and the observation showed that it is a Gram-positive bacteria. The morphological characteristics of WHPC005 colonies on the LB plate were convex, with mucous, and with a smooth edge (Figure 2A). The cellular morphology of WHPC005 was further identified and confirmed as *Sphaerita* through scanning electron microscopy (Figure 2B).

Moreover, the sequencing analysis indicated that WHPC005 shared 99% homology with *Lactosphaera pasteurii* by comparing the 16S rRNA sequences using the BLAST data package available on the NCBI website. A phylogenetic tree was constructed based on sequences of the 16S rRNA gene (Figure 3). WHPC005 shared the highest genetic relationship with *Trichococcus pasteurii* strain ATCC 35945, which belonged to genera of lactic acid bacteria, and in contrast, had the farest relationship with *Wolbachia endosymbiont* strain T2. Therefore, WHPC005 was thus verified as a strain of *Lactosphaera pasteurii* that produced the highest amount of α-galactosidase based on morphological and molecular evidence, and was designated as *L. pasteurii* WHPC005.

### 3.3. Effects of Temperature and pH on α-Glucosidase Activity from L. pasteurii WHPC005

To determine the effects of temperature on the α-galactosidase activity from *L. pasteurii* WHPC005, a series of different induction temperatures was tested, and the data suggested that 45 °C was the optimum temperature of α-galactosidase for *L. pasteurii* WHPC005 (Figure 4A).

In addition, the initial pH of the medium ranged from 2.5 to 8.0, and was adjusted to further evaluate the effects of pH on the productivity of α-galactosidase. The results showed that the highest enzymatic activity was obtained at pH 5.5 (Figure 4B).

### 3.4. Effects of Metal Ions and Chemical Reagents on α-Glucosidase Activity of L. pasteurii WHPC005

The sensitivity of α-galactosidase from *L. pasteurii* WHPC005 to various concentrations of metal ions was tested. The results are summarized in Table 1. The activity of α-galactosidase was significantly inhibited in the presence of K^+^ (7–23% residual activity) and Al^3+^ (0–12% residual activity), and was completely inhibited by Fe^3+^ (0% residual activity). In contrast, Ca^2+^ (110–146% residual activity), Fe^2+^ (128–155% residual activity), Mn^2+^ (109–140% residual activity), and Zn^2+^ (126–158% residual activity) significantly enhanced the activity of α-galactosidase among the tested concentrations. Mg^2+^ and Cu^2+^ inhibited α-galactosidase at the concentrations of 10 mM and 5 mM, whereas the concentration of 1 mM enhanced the α-galactosidase activity.

The sensitivity of α-galactosidase from *L. pasteurii* WHPC005 to various concentrations of chemical reagents was confirmed. The results are listed in Table 2. For the tested sugars, fructose (96.50–98.50% residual activity), sucrose (90.48–98.62% residual activity), lactose (86.64–96.38% residual activity), and galactose (32.76–94.52% residual activity) inhibited α-galactosidase. Moreover, when the galactose concentration was 100 mM was the inhibitory effect was very strong. On the contrary, the presence of glucose and raffinose enhanced the activity of α-galactosidase.

It should be noted that the inhibited or enhanced effects of sugars were concentration-dependent. As a strong denaturing reagent that disrupts the steric structures of proteins, SDS brought about a 84.32% to 94.76% loss of activity of α-galactosidase when the concentration of SDS was increased from 10 to 100 mM. Due to the disintegration of the tertiary structure, decreased activity was found in the presence of the chelating agent EDTA (85.56–91.23% residual activity), suggesting that the α-galactosidase from the WHPC005 strain is a metalloenzyme. NaCl (76.11% residual activity) and (NH_4_)_2_SO_4_ (87.16% residual activity) at 100 mM concentration had little effect on the α-galactosidase, which may be the consequence of a salt-stabilizing effect [13].

### 3.5. Induction and Expression of α-Galactosidase from L. pasteurii WHPC005

To further confirm the induced effect of α-galactosidase by raffinose, we measured the enzymatic activity with different concentrations of raffinose and glucose. The data indicated that the expression of α-galactosidase could be induced by raffinose and glucose (Figure 5A). However, there was a stronger induced effect for raffinose than for glucose, and the maximum induced effect for raffinose occurred at a concentration of 30 μmol/L (Figure 5A,B). We also analyzed the α-galactosidase expression induced by raffinose (30 μmol/L) at different times, and the results showed that the molecular size of the α-galactosidase in WHPC005 was approximately 50 kD. The α-galactosidase was strongly induced by raffinose after 2 h of treatment, and a large accumulation resulted after 6 h of treatment (Figure 5C).

### 3.6. Cloning of the α-Galactosidase Gene from L. pasteurii WHPC005 and Its Sequence Analysis

The genomic DNA extracted from *L. pasteurii* WHPC005 was digested with different concentrations of *Sau*3A I (0.2, 0.4, 0.6, and 0.8 U/µL) to optimize the random digestion. As shown in Figure 6A, the optimal reaction conditions were 1 μg of genomic DNA digested with 0.8 U of *Sau*3A I for 120 min to obtain random genomic fragments 3–5 kb in length. These random fragments were ligated into the pUC19 plasmid through the *Bam*H I site, and then transferred into competent DH5α cells. The positive clones were identified with X-α-gal (20 µg/mL) (Figure 6B).

Sequence analysis demonstrated that the cloned DNA fragment was 3039 bp in total and covered the entire α-galactosidase gene and its promoter. The α-galactosidase gene was named *LpagaH*. Further analysis indicated that the open reading frame of the *LpagaH* gene was 1230 bp, which encoded 409 putative amino acids (Appendix A).

### 3.7. Bioinformatics Analysis of the α-Galactosidase from L. pasteurii WHPC005

The results of the prediction analysis showed that the isoelectric point (pI) of α-galactosidase from *L. pasteurii* WHPC005 was 4.84, and that its molecular weight (MW) was 47.40 kD. Prediction analysis also demonstrated that this α-galactosidase contained cAMP- and CGMP-dependent protein kinase phosphorylation sites, protein kinase C-terminal phosphorylation sites, type II casein kinase phosphorylation sites, tyrosine kinase phosphorylation sites, and amination sites.

The secondary structure prediction results indicated that this α-galactosidase was rich in random coils (55.5%), and also contained alpha helixes (28.58%) and beta turns (14.91%) (Appendix A). In addition, the results of the transmembrane region prediction analysis showed that this α-galactosidase was not a transmembrane protein and had no transmembrane region structure (Appendix A).

Understanding and analyzing the tertiary structure of the protein is helpful for clarifying the relationship between protein structure and function. The tertiary structure of α-galactosidase from *L. pasteurii* WHPC005 was predicted, and the results are displayed in Figure 7. The arrow indicates the highly conserved region of this α-galactosidase when compared to other α-galactosidases, and highly variable regions are shown in blue (Figure 7A). Alpha helixes and beta turns were also found in the conserved region of this α-galactosidase (Figure 7B).

## 4. Discussion

Microorganisms are ideal sources of α-galactosidases due to their high expression level, short culture time, and extracellular secretion [14]. The production and enzymic properties of the α-galactosidases from various bacteria and fungus, such as *Bacillus subtilis* [15], *Rhizomucor miehei* [16], and *Bacteroides thetaiotaomicron* [17], have been widely studied and reported. However, reports on the α-galactosidases and their enzymatic characteristics from *L. pasteurii* remain rare.

Herein, a bacterial strain (numbered as WHPC005) with a high level of α-galactosidases production was obtained and subsequently identified as *L. pasteurii*. The enzymatic characteristic data indicated that the optimum temperature and pH for *L. pasteurii* WHPC005 were 45 °C and 5.5, respectively. Wang et al. purified α-galactosidases from *Neosartorya fischeri* with a higher optimum temperature of 60 °C to 70 °C, and a lower pH of 4.0, as compared to *L. pasteurii* WHPC005 [10]. Moreover, a novel α-galactosidase was purified and characterized from *Bacillus megaterium*, and its high specific activity at 37 °C and pH 6.8, which were parameters that were lower and higher than those of *L. pasteurii* WHPC00, respectively [18]. In addition, Chen et al. recombined the α-galactosidases from *Rhizomucor miehei* in *Pichia pastoris* [16]. The optimum temperature of this recombinant enzyme was 55 °C, which was higher than that of *L. pasteurii* WHPC005, and with the same optimum pH. Therefore, the optimum temperature and pH of α-galactosidases from different microorganism species are different.

It should be noted that the effects of metal ions on the activity of α-galactosidase were similar between *Pleurotus djamor* and *L. pasteurii*, whereas the effects of chemical reagents on the α-galactosidase activities of these two species had some different effects; for example, glucose increased the α-galactosidase activity for *L. pasteurii,* but had no effect for *P. djamor* (Table 1 and Table 2).

There are 11 genera of Gram-positive lactic acid bacteria (LAB), such as *Lactococcus, Streptococcus, Leuconostoc, Pediococcus,* and *Carnobacterium* [19]. In our research, we obtained an α-galactosidase-producing strain of lactic acid bacteria that was isolated from soil in the XiangXi autonomous prefecture. Its enzymatic activity can be induced in the presence of glucose and raffinose. However, there was a stronger induced effect by raffinose as compared to glucose. Raffinose, also called melitriose, is a trisaccharide with galactose residues joined by α-1,6-glycosidic bonds to sucrose and hydrolyzed by α-galactosidase [20]. Although the expression of α-galactosidase can be induced by raffinose, the optimum concentrations of raffinose required to induce α-galactosidase expression for different microbial species are different. We found that the optimum concentration of raffinose for *L. pasteurii* WHPC005 induction was 30 μmol/L, which was different from that for *Thermus thermophilus* HB27 [21], *Bifidobacterium longum* JCM 7052 [22], and *Bacillus coagulans* NRR1207 [23].

In *Streptococcus mutans*, the α-galactosidase gene is associated with a multiple-sugar metabolism operon [24]. This α-galactosidase gene is essential for growth when melibiose and raffinose exist. In the Gram-positive bacterium *Carnobacterium piscicola,* the α-galactosidase determinant is associated with two α-galactosidase genes, and both enzymatic activities are repressed in the presence of glucose and induced in the presence of melibiose or raffinose [25,26,27]. In the current study, we also cloned the α-galactosidase gene and performed a bioinformatics predictive analysis, which was helpful for further understanding the function of this α-galactosidase from *L. pasteurii* WHPC005.

In this study, we only checked the α-galactosidase expression by SDS-PAGE. Pure α-galactosidase will be separated from this strain of *L. pasteurii* through the column chromatographic separation method, and the enzyme gene will also be modified via gene engineering to increase the activity of α-galactosidase in the future, which will expand its potential application in the feed industry, sugar production, and the pharmaceutical industry.

## 5. Conclusions

To the best of our knowledge, this study is the first to systematically analyze an α-galactosidase-producing *Lactosphaera pasteurii* strain, and its enzymatic expression analysis. The optimal temperature and pH for this α-galactosidase were 45 °C and 5.5, respectively. The activity of α-galactosidase was inhibited by K^+^, Al^3+^, Fe^3+^, fructose, sucrose, lactose, galactose, SDS, EDTA, NaCl, and (NH_4_)_2_SO_4_, and enhanced by Ca^2+^, Fe^2+^, Mn^2^, Zn^2+^, glucose, and raffinose. The optimal inducer was raffinose, and the optimal induction concentration was 30 μmol/L. The open reading frame of the α-galactosidase gene was 1230 bp, which encodes a putative protein of 409 amino acids in length. The isoelectric point and molecular weight of this α-galactosidase were 4.84 and 47.40 kD, respectively. Random coils, alpha helixes, and beta turns were observed in its secondary structure, and conserved regions were found in the tertiary structure of this α-galactosidase. Therefore, this α-galactosidase-producing bacterial strain has the potential for application in the feed industry.

## Figures and Tables

**Figure 1 molecules-27-05942-f001:**
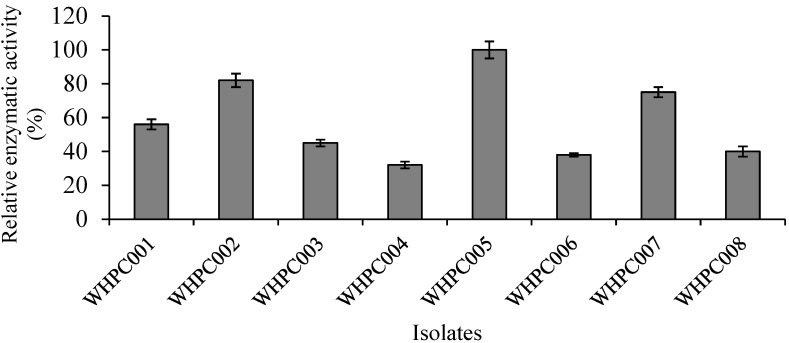
Eight colonies (isolated from the soil near a tofu factory) producing α-galactosidase.

**Figure 2 molecules-27-05942-f002:**
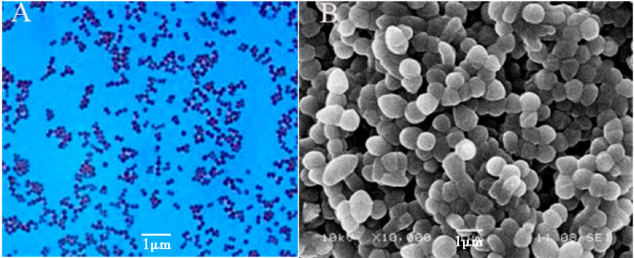
Phenotypic characterization assay of the isolated strains. (**A**) Gram staining of the isolated strains (400×); (**B**) scanning electron micrograph of the isolated strains (1000×). Bar = 1 μm.

**Figure 3 molecules-27-05942-f003:**
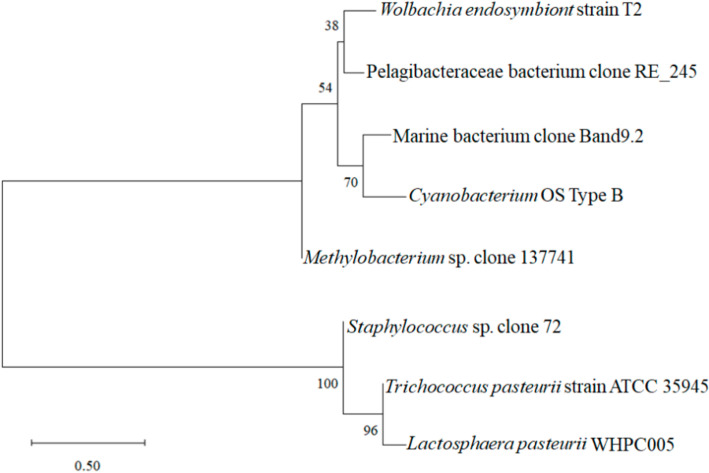
Phylogenetic tree based on the 16S rDNA gene sequences of *L. pasteurii* WHPC005.

**Figure 4 molecules-27-05942-f004:**
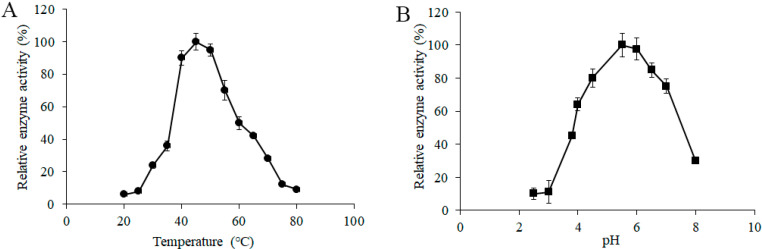
Effect of (**A**) temperature and (**B**) pH on the α-galactosidase activity from *L. pasteurii* WHPC005.

**Figure 5 molecules-27-05942-f005:**
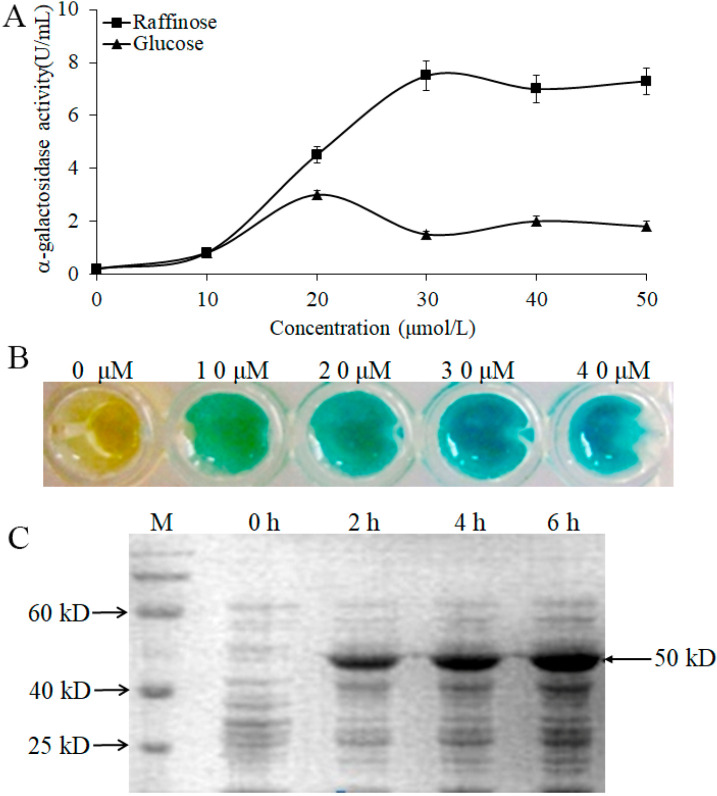
Induction expression of the α-galactosidase from *L. pasteurii* WHPC005 by raffinose. (**A**) α-Galactosidase activity with different induced concentrations of raffinose and glucose. (**B**) Induction effects with different concentrations of raffinose as indicated by the color reaction. (**C**) Induction of α-galactosidase expression with 30 μmol/L raffinose at various times (0 h, 2 h, 4 h, and 6 h) via SDS-PAGE.

**Figure 6 molecules-27-05942-f006:**
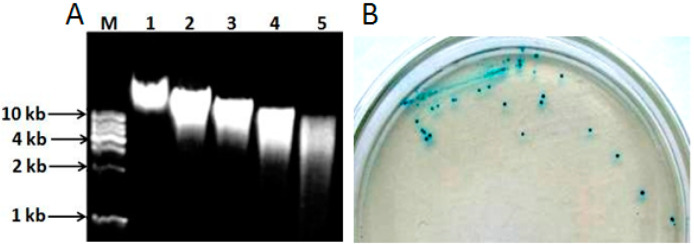
Cloning of the α-galactosidase gene from *L. pasteurii* WHPC005. (**A**) Genomic DNA digested using diluted *Sau*3AI in 0.8% agarose gel. Lane M: 1 kb DNA ladder; Lane 1: Genomic DNA without digestion; Lanes 2–5: digestion for 120 min with 0.2 U, 0.4 U, 0.6 U, or 0.8 U *Sau*3AI enzyme, respectively. (**B**) Positive clones were identified on X-α-gal medium via color reaction.

**Figure 7 molecules-27-05942-f007:**
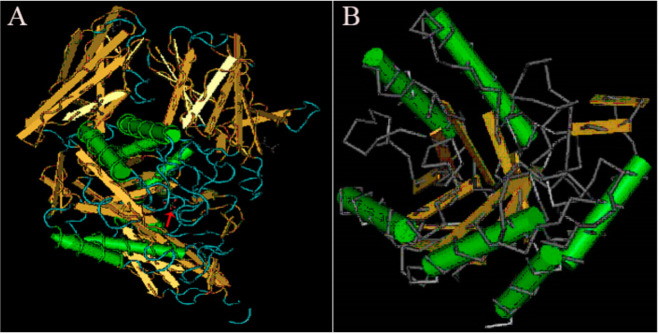
Results of the tertiary structure prediction for the α-galactosidase from *L. pasteurii* WHPC005. (**A**) Overall tertiary structure of the α-galactosidase. (**B**) Conserved tertiary structure of the α-galactosidase.

**Table 1 molecules-27-05942-t001:** Effects of metal ions on the activity of α-galactosidase from *L. pasteurii* WHPC005.

Metal Ion Concentration	Relative α-Galactosidase Activity (%)		
	10 mM	5 mM	1 mM
K^+^	17 ± 0.05	7 ± 0.04	23 ± 0.08
Ca^2+^	110 ± 0.50	146 ± 0.80	132 ± 1.20
Fe^2+^	155 ± 0.60	146 ± 0.40	128 ± 0.00
Mg^2+^	80 ± 0.30	94 ± 0.20	122 ± 0.40
Mn^2+^	109 ± 0.60	140 ± 1.20	114 ± 0.70
Zn^2+^	158 ± 1.60	142 ± 0.70	126 ± 0.80
Cu^2+^	10 ± 0.02	8 ± 0.00	110 ± 0. 40
Al^3+^	0 ± 0.00	0 ± 0.01	12 ± 0.05
Fe^3+^	0 ± 0.00	0 ± 0.00	0 ± 0.00

**Table 2 molecules-27-05942-t002:** Effects of chemical reagents on the activity of α-galactosidase from *L. pasteurii* WHPC005.

Effector Concentration	Relative α-Galactosidase Activity (%)		
	100 mM	50 mM	10 mM
Glucose	104 ± 0.10	136 ± 0.42	112 ± 0.30
Fructose	96.50 ± 0.05	98.4 ± 0.03	98.5 ± 0.08
Sucrose	90.48 ± 0.04	97.86 ± 0.13	98.62 ± 0.09
Lactose	86.64 ± 0.02	92.11 ± 0.12	96.38 ± 0.56
Galactose	32.76 ± 0.01	78.15 ± 0.24	94.52 ± 0.47
Raffinose	156.44 ± 0.48	192.86 ± 1.36	120.47 ± 0.35
SDS	5.24 ± 0.12	7.65 ± 0.24	15.68 ± 0.12
EDTA	85.56 ± 0.24	87.39 ± 0.58	91.23 ± 0.45
NaCl	76.11 ± 0.72	92.85 ± 0.50	94.37 ± 0.69
(NH_4_)_2_SO_4_	87.16 ± 0.11	95.96 ± 0.35	98.44 ± 0.12

## Data Availability

Not applicable.

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
