# Peer review of "Isolation and Identification of an α-Galactosidase-Producing Lactosphaera pasteurii Strain and Its Enzymatic Expression Analysis"

_molecules, 2022, doi:10.3390/molecules27185942_

Round 1

Reviewer 1 Report

The research is well conceived and executed and described in appropriate detail. Please respond to the following.

1. The approach to isolation “collected from locations rich in soybean meal” is logical. Was this approach successful in the isolation of an enzyme withproperties well adapted to the anticipated uses? Alternatively, is this simply another enzyme without properties distinct from others described?

2. The general reader may not be familiar with the phylogenetic assignment for the species identified in Figure 3. Please expand upon the briefdescription (There are 11 genera of gram-positive lactic acid bacteria…) and provide additional details as to the placement of Lactosphaera pasteurii inthe overall categories of procaryotes, eucaryotes, and related sub-categories.

3. The characterization of the enzyme properties in Figures 4-5 and Tables 1-2 is appropriate. Please provide a detailed comparison of theseproperties with the enzymes from other sources. Are any of the properties of this new enzyme unique or exceptional? Are there properties that suggestadvantages in specific applications such as medical or industrial? If so, which applications?

4. How many enzymes of this class have been described? What are the characteristics of this new enzyme or the method of isolation that justify a fullmanuscript?

Author Response

Dear Editor:

This is a revision of our previous manuscript (molecules-1909935). First, we thank the reviewers for their critical comments and constructive suggestions. Based on the suggestions, we performed many revisions of this manuscript in red color. For the response to reviewers, the comments are black, author's answers are blue. In addition, our manuscript was also edited for English language by LetPub, and the certificate of English language editing will be uploaded to the manuscript system.

Reviewer 1:

Comments and Suggestions for Authors

The research is well conceived and executed and described in appropriate detail. Please respond to the following.

  1. The approach to isolation “collected from locations rich in soybean meal” is logical. Was this approach successful in the isolation of an enzyme with properties well adapted to the anticipated uses? Alternatively, is this simply another enzyme without properties distinct from others described?

Answer: In this manuscript we described the properties of the α-galactoside of Lactosphaera pasteurii including effects of temperature, pH, metal ions and chemical reagents on the enzymatic activity. In fact, other properties of the α-galactoside, such as dynamic changes of enzymatic activity during fermentation, and its proteinase activity were also checked. Its enzymatic gene was also modified by site-directed mutation method to improve its enzymatic activity. These results we did not list in this manuscript.

  1. The general reader may not be familiar with the phylogenetic assignment for the species identified in Figure 3. Please expand upon the brief description (There are 11 genera of gram-positive lactic acid bacteria…) and provide additional details as to the placement of Lactosphaera pasteurii inthe overall categories of procaryotes, eucaryotes, and related sub-categories.

Answer: We re-constructed the phylogenetic tree, and described the genetic relationship with other species. 

  1. The characterization of the enzyme properties in Figures 4-5 and Tables 1-2 is appropriate. Please provide a detailed comparison of these properties with the enzymes from other sources. Are any of the properties of this new enzyme unique or exceptional? Are there properties that suggestadvantages in specific applications such as medical or industrial? If so, which applications?

Answer: We had corrected it.

  1. How many enzymes of this class have been described? What are the characteristics of this new enzyme or the method of isolation that justify a full manuscript?

Answer:Microorganisms are ideal sources of α-galactosidases due to their high expression level, short culture time, and extracellular secretion. The production and enzymic properties of the α-galactosidases from various bacteria and fungus, such as Bacillus subtilis, Rhizomucor miehei, and Bacteroides thetaiotaomicron, have been widely studied and reported. We will compare the enzyme properties from these different species in other manuscript in the future.

Sincerely yours,

Professor Liu

Guizhou Institute of Technology, Guiyang, China

Reviewer 2 Report

In this manuscript, the author described the identification of a strain of Lactosphaera pasteurii (WHPC005) that produces α-galactosidase. The phenotype of WHPC005 was analyzed using Gram staining and scanning electron micrograph. α-Glucosidase activity and the gene from L. pasteurii WHPC005 were analyzed and cloned. They also conducted the tertiary structure prediction for the α-galactosidase from L. pasteurii WHPC005. The results potentially adds to our knowledge of α-galactosidase in Lactosphaera pasteurii but are shrouded in doubt by lots of issues in this manuscript as explained below.

Major Concerns:

1) There is no standard errors in the data analysis, such as Figure 4, Figure 5A.

2) The CDS shoul be cloned and shown in the form of agarose gel electrophoresis.

Minor Concerns:

1) In line 21, 409 should be 408.

2) In line 44, Ug/L was not suitable.

3) In line 69, the formation of reference doesn’t meet the requirements.

4) There was no scale bar in the Figure 2.

5) In Table, there were the unnecessary space and these spaces should be removed.  

Author Response

Dear Editor:

This is a revision of our previous manuscript (molecules-1909935). First, we thank the reviewers for their critical comments and constructive suggestions. Based on the suggestions, we performed many revisions of this manuscript in red color. For the response to reviewers, the comments are black, author's answers are blue. In addition, our manuscript was also edited for English language by LetPub, and the certificate of English language editing will be uploaded to the manuscript system.

Reviewer 2:

Comments and Suggestions for Authors

In this manuscript, the author described the identification of a strain of Lactosphaera pasteurii (WHPC005) that produces α-galactosidase. The phenotype of WHPC005 was analyzed using Gram staining and scanning electron micrograph. α-Glucosidase activity and the gene from L. pasteurii WHPC005 were analyzed and cloned. They also conducted the tertiary structure prediction for the α-galactosidase from L. pasteurii WHPC005. The results potentially adds to our knowledge of α-galactosidase in Lactosphaera pasteurii but are shrouded in doubt by lots of issues in this manuscript as explained below.

Major Concerns:

1) There is no standard errors in the data analysis, such as Figure 4, Figure 5A.

Answer: We had corrected it.

2) The CDS should be cloned and shown in the form of agarose gel electrophoresis.

Minor Concerns:

Answer: We had shown the sequences of the CDS in supplementary materials of this manuscript, it is unnecessary to add the agarose gel electrophoresis result.

1) In line 21, 409 should be 408.

Answer: The open reading frame of the α-galactosidase gene was 1230 bp, which encodes a putative protein of 409 amino acids in length.

2) In line 44, Ug/L was not suitable.

Answer: We had corrected it.

3) In line 69, the formation of reference doesn’t meet the requirements.

Answer: We had corrected it.

4) There was no scale bar in the Figure 2.

Answer: We had corrected it.

5) In Table, there were the unnecessary space and these spaces should be removed.

Answer: We are sorry that we can’t find the unnecessary space in table.

Sincerely yours,

Professor Liu

Guizhou Institute of Technology, Guiyang, China

Reviewer 3 Report

Dear authors, kindly perform the minor revision as follows:
I recommend a minor revision
. Kindly look into the following comments to improve the manuscript:

Overall, this study doesn’t present the novelty finding, thus please highlight the contribution of your study to the research progress in this field. As authors mentioned in Conclusion (page 10 line 306-307) about the potential for application in the feed industry, please add more discussions how this research can fill into the current research gap?

Materials and Methods

1.       Methodology of enzyme production is missed. Please described more details of the enzyme used for characterization in this study. Is it a crude or partial purified enzyme? How to produce and harvest? What condition used to produce?

2.       Please correct the pH range. Is it 2.6-8.0 (Materials and Methods, page 2 line 88) or 2.5-8.0 (Results and Analysis, page 5 line 161)?

Results and Analysis

1.       Data of relative enzyme activity at 20 and 80°C is missed in Figure 4A).    

2.       Corrected M in Figure 5B. Should it be mM (mmol/L)?

3.       In Figure 5C, is it SDS-PAGE or agarose? Please clearly describe.

4.       ORF of gene composed of 1230 bp (page 8 line 229) or 1227 bp (page 10 line 301)?

Discussion

Please mention the limitation of your study and suggest some possible works in the future. Add more information (literature search) to explain how benefit of this work.

Author Response

Dear Editor:

This is a revision of our previous manuscript (molecules-1909935). First, we thank the reviewers for their critical comments and constructive suggestions. Based on the suggestions, we performed many revisions of this manuscript in red color. For the response to reviewers, the comments are black, author's answers are blue. In addition, our manuscript was also edited for English language by LetPub, and the certificate of English language editing will be uploaded to the manuscript system.

Reviewer3:

Comments and Suggestions for Authors

Dear authors, kindly perform the minor revision as follows:

I recommend a minor revision. Kindly look into the following comments to improve the manuscript:

Overall, this study doesn’t present the novelty finding, thus please highlight the contribution of your study to the research progress in this field. As authors mentioned in Conclusion (page 10 line 306-307) about the potential for application in the feed industry, please add more discussions how this research can fill into the current research gap?

Materials and Methods

  1. Methodology of enzyme production is missed. Please described more details of the enzyme used for characterization in this study. Is it a crude or partial purified enzyme? How to produce and harvest? What condition used to produce?

Answer: We had corrected it.

  1. Please correct the pH range. Is it 2.6-8.0 (Materials and Methods, page 2 line 88) or 2.5-8.0 (Results and Analysis, page 5 line 161)?

Answer: We had corrected it.

Results and Analysis

  1. Data of relative enzyme activity at 20 and 80°C is missed in Figure 4A).

Answer: We had corrected it.

  1. Corrected M in Figure 5B. Should it be mM (mmol/L)?

Answer: We had corrected it.

  1. In Figure 5C, is it SDS-PAGE or agarose? Please clearly describe.

Answer: We had corrected it.

  1. ORF of gene composed of 1230 bp (page 8 line 229) or 1227 bp (page 10 line 301)?

Answer: We had corrected it.

Discussion

Please mention the limitation of your study and suggest some possible works in the future. Add more information (literature search) to explain how benefit of this work.

Answer: We had corrected it.

Sincerely yours,

Professor Liu

Guizhou Institute of Technology, Guiyang, China

Round 2

Reviewer 2 Report

The research strategy is well oriented and provides new and interesting information about the α-Galactosidase in Lactosphaera pasteurii Strain. I accept the current manuscript for publication.